Unleashing quantum algorithms with Qinterpreter: bridging the gap between theory and practice across leading quantum computing platforms

Contreras-Sepúlveda Wilmer 1 wilmer.contreras@inaoep.mx
Villegas-Martínez Braulio Misael 2 braulio.villegas@uaem.mx
http://orcid.org/0000-0002-6051-0673 Gesing Sandra 3
Sánchez-Mondragón José Javier 1
http://orcid.org/0009-0008-7008-8534 Sánchez-Pérez Juan Carlos 4
Vidales-Basurto Claudia Andrea 5
Escobedo-Alatorre J. Jesús 2
Torres-Palencia Angel David 1
Palillero-Sandoval Omar 2
Licea-Rodriguez Jacob 2
Lozano-Crisóstomo Néstor 6
García-Melgarejo Julio César 6
Palacios-Perez Eddie Nelson 7
1 Instituto Nacional de Astrofísica, Óptica y Electrónica , Tonantzintla, Puebla , Mexico
2 Centro de Investigación en Ingeniería y Ciencias Aplicadas, Universidad Autónoma del Estado de Morelos , Cuernavaca , México
3 San Diego Supercomputer Center, University of California, San Diego , La Jolla, CA , United States
4 Facultad de Ciencias Físico-Matemáticas, Benemérita Universidad Autónoma de Puebla , Heroica Puebla de Zaragoza, Puebla , México
5 Centro de Investigación en Matemáticas A.C. Jalisco S/N, Col. Valenciana , Guanajuato, Guanajuato , Mexico
6 Facultad de Ingeniería Mecánica y Eléctrica, Universidad Autónoma de Coahuila , Torreón, Coahuila , México
7 Centro Regional de Radioterapia Zona Norte , Ciudad Juárez, Chihuahua , México
Cherukuri Aswani Kumar
Electronic publication date: 2024 Oct 15
Publication date: 2024
Volume: 10
Electronic Location ID: e2318
Received 2023 Sep 7; Accepted 2024 Aug 20
Copyright: © 2024 Contreras-Sepúlveda et al.
Copyright year: 2024
Copyright holder: Contreras-Sepúlveda et al.
License: This is an open access article distributed under the terms of the Creative Commons Attribution License, which permits unrestricted use, distribution, reproduction and adaptation in any medium and for any purpose provided that it is properly attributed. For attribution, the original author(s), title, publication source (PeerJ Computer Science) and either DOI or URL of the article must be cited.
License URL: https://creativecommons.org/licenses/by/4.0/

Keywords: Education tool, Interpreter, Quantum computing

Funding: The authors received no funding for this work.

==============================
Quantum computing is a rapidly emerging and promising field with the potential to transform various research domains including drug design, network technologies, and sustainable energy solutions. Due to the inherent complexity and divergence from classical computing, several major quantum computing libraries have been developed to implement quantum algorithms, namely IBM Qiskit, Amazon Braket, Cirq, PyQuil, and PennyLane. These libraries enable quantum simulations on classical computers and execution on corresponding quantum hardware, such as Qiskit programs on IBM quantum computers. Despite the variations among these platforms, the core concepts remain the same. One notable challenge is the absence of a Python-based quantum interpreter to connect these five frameworks, a gap that remains to be fully addressed. In response, our work introduces a tool called Qinterpreter, accessible through a user-friendly web interface, the Quantum Science Gateway QubitHub, which operates alongside Jupyter Notebooks. Built using the Python Object-Oriented Programming System, Qinterpreter unifies the five well-known quantum libraries into a single framework. Designed as an educational tool for students and researchers entering the quantum domain, Qinterpreter enables the straightforward development and execution of quantum circuits across such platforms. This work highlights the quantum programming versatility and accessibility of Qinterpreter and underscores our ultimate goal of pervading Quantum Computing through younger, less specialized, and diverse cultural and national communities.

Introduction

Quantum computing has emerged as a burgeoning area at the intersection of physics and computer science, offering the groundbreaking potential for solving problems beyond classical computations’ scope. At the core of this revolution are quantum bits, or qubits, which represent physical quantum systems existing between two distinct states, such as the spin of an electron (Harneit, 2002; Pla et al., 2012), the polarization of a photon (Zhong et al., 2020; Wang et al., 2020; O’Brien, 2007; O’Brien, Furusawa & Vučković, 2009), or the energy states of an atom (Briegel et al., 2000; Deutsch, Brennen & Jessen, 2000; Saffman, 2016). Unlike classical bits, which are limited to a 0 or 1 state, qubits can exist simultaneously in all possible configurations of both states due to a superposition phenomenon (Dirac, 1981). This unique quantum property, along with the principles of entanglement (Nielsen & Chuang, 2010) and interference (Griffiths & Schroeter, 2018), bestows quantum computers with a significant advantage, enabling them to address specific computational challenges with greater efficiency and speed than their classical counterparts. By leveraging the power of atoms and photons (Landauer, 1991), quantum computing holds promise for diverse applications, including mitigating cyber threats posed by nation-state actors to quantum-safe encryption (Shor, 1999; Grover, 1996), biomanufacturing (Andersson et al., 2022; Bernal et al., 2022), and quantum artificial intelligence (Dunjko & Briegel, 2018; Wichert, 2020).

To propel the field of quantum computing forward, the development of platforms and libraries that enable the creation of quantum programs for cutting-edge hardware is of the utmost importance. Industry leaders such as IBM, Amazon, Google, Rigetti Computing, and Xanadu are actively developing their open-source programming languages and libraries, like Qiskit, Amazon Braket, Cirq, PyQuil, and PennyLane (Abraham et al., 2019; Gonzalez, 2021; Pattanayak, 2021; pyQuil, http://pyquil.readthedocs.io/en/latest; Xanadu Hardware, https://www.xanadu.ai/hardware/; Rigetti Computing, https://www.rigetti.com/systems; Google Quantum, https://research.google/teams/applied-science; Fortunato, Campos & Abreu, 2022; Cirque Developers, 2022; Bergholm et al., 2018; Amazon Web Services, 2020). Notably, these languages, primarily based on Python code, are specifically designed to describe the creation, manipulation, and execution of quantum circuits and operations. Additionally, they provide valuable tools to facilitate a comprehensive understanding of the fundamental principles of quantum computing. Nevertheless, it is essential to acknowledge that while these companies offer cloud-based access to quantum computing resources, acquiring expertise to wield these resources effectively can be challenging for newcomers and those unfamiliar with the field. This potential difficulty may hinder individuals from fully leveraging these invaluable resources. Additionally, it’s noteworthy that IBM’s Qiskit, Amazon’s Braket, Google’s Cirq, Rigetti’s PyQuil, and Xanadu’s PennyLane are Python-based, open-source quantum computing libraries serving as interfaces, compilers, and execution environments for quantum programs on computers or simulators. Some of these libraries incorporate connectors to promote cross-framework compatibility. For instance, Qiskit users can use the qBraid SDK to transpile circuits with Braket (qBraid, https://www.qbraid.com/), and Amazon Braket integrates PennyLane and Qiskit through plugins (Amazon Web Services, 2020). PennyLane supports circuit imports from these libraries via its plugins, allowing native programming (Xanadu Hardware, Toronto, Canada). However, it is crucial to highlight that none of these libraries possess the capability to directly translate code from one platform to another. This limitation needs a complete reprogramming of algorithms rather than a simple translation process. This stands in contrast to classical computing, where interpreters like Python and Ruby adeptly manage diverse processors, executing user instructions without separate compilation steps (Ancona et al., 2007; Sanner, 1999). These interpreters can work with different kinds of libraries, where the interpreter must comprehend and manage the library’s code and simplify programming for non-professionals (Ancona et al., 2007; Barrett, Bolz & Tratt, 2015; Hutton, 1993; Thomas & Hunt, 2001; Beasley, 2006; Sanner, 1999; Power & Rubinsteyn, 2013; Gutschnidt, 2003; van Rossum & Drake, 2004; Østerlie, 2002; Matsumoto & Ishituka, 2002; Harris et al., 2020; Van Der Walt, Colbert & Varoquaux, 2011). Recognizing the challenge posed by the absence of direct code translation in the quantum domain, we introduce Qinterpreter, a quantum interpreter hosted on the Qubithub platform (www.qubithub.org). This quantum interpreter is a pilot and will be introduced to the Latin American quantum community via the successful Latin America Optics and Photonics Workshops conducted since 2010 (Sanchez-Mondragon, 2020). International partners can also gain access by directly contacting the platform’s contact form.

Qinterpreter is a library that combines the most popular quantum computing libraries—Qiskit, Pyquil, Pennylane, Amazon-Braket, and Cirq. While existing open-source software like Mitiq (LaRose et al., 2022) partially approaches this intended goal, it primarily focuses on error mitigation techniques for noisy quantum computers, not serving as a general quantum programming language or interpreter. Unlike Mitiq, Qinterpreter consolidates the mentioned libraries into a unified framework, enabling interaction and code execution across all five quantum computing platforms. Practically, Qinterpreter acts as a simulator that translates algorithms between these frameworks and is freely accessible online (Contreras Sepulveda & Contributors, 2023). This proves beneficial, as quantum computing simulators are crucial for researchers and newcomers who lack access to a physical quantum computer. These simulators allow the development and testing of quantum algorithms without hardware constraints. They can calculate the outcomes of applying a quantum circuit in a user-friendly way without the need for installations and configurations on the user side (Altman et al., 2021). Hence, the users can employ Qinterpreter on classical computers to simulate local quantum processors or quantum simulators, facilitating the execution of quantum gates and measurements on each local quantum processor, adhering to the unified library’s rules. This unified approach empowers individuals at all levels of expertise, from beginners to advanced, to effectively use the implemented algorithms within the Qinterpreter language.

It’s crucial to mention that the optimization techniques for quantum gates and defining other unitary gates from a universal set of gates are beyond the scope of this work, as they are well-established within the field of quantum computing. Furthermore, Qinterpreter distinguishes itself through its foundation in the Python Object-Oriented Programming System (OOPs), a programming paradigm rooted in object concepts (Ramkarthik & Barkataki, 2022). This execution is further supported by dedicated Jupyter Notebooks for each code, and its uniqueness lies in its technology-agnostic approach, which can be used as a backend to other quantum computing frameworks based on Python. This means that Qinterpreter’s structure allows any Python-accessible quantum computer simulator to serve as a backend, leveraging existing simulators from all quantum libraries involved. It is worth mentioning that, to the best of our knowledge, no equivalent platform exists to the Qinterpreter at present.

This work is organized as follows: “How Qinterpreter Works” provides an overview of the requirements and installation instructions for Qinterpreter, supported by an explanation of its functionalities. To assess the performance of Qinterpreter across five frameworks—Qiskit, Amazon Braket, Cirq, PyQuil, and PennyLane—, we reproduce three widely recognized quantum computing examples: the generation of one of the four Bell’s states, commonly known as Bell state 00, Grover’s algorithm and the benchmark problem for factoring 15, 21 and 35 by using the Shor’s algorithm. “Application of the Qinterpreter” delves into these examples, providing step-by-step instructions and explanations of the Qinterpreter’s functions in handling these circuits. Lastly, “Conclusions and Future Work” wraps the work with concluding remarks and insightful observations.

How qinterpreter works

The Qinterpreter operations are structured into three sequential steps. The initial step involves generating a common language, specifically Qinterpreter. The second step consists of translating this language into five distinct frameworks: Qiskit, Cirq, PennyLane, PyQuil, and Amazon Braket; while the last step centers on managing the simulation process. With this in mind, Qinterpreter aims to offer an open-source environment that enables users to interact with quantum backends. These backends serve as interfaces to quantum simulators, allowing users to engage in simulation or execution without underlying in-depth knowledge of the technical details. The main file, main.py in Contreras Sepulveda & Contributors (2023), is the central interface for translation and simulation

# main.py from quantumgateway.quantum_translator.braket_translator import BraketTranslator from quantumgateway.quantum_translator.cirq_translator import CirqTranslator from quantumgateway.quantum_translator.qiskit_translator import QiskitTranslator from quantumgateway.quantum_translator.pennylane_translator import PennyLaneTranslator from quantumgateway.quantum_translator.pyquil_translator import PyQuilTranslator

In this scenario, the code imports essential modules and classes crucial for quantum computing, including translators tailored to the five distinct libraries. Each translator class, such as braket_translator, cirq_translator, etc., is implemented in separate files within the quantumgateway.quantum_translator package, which acts as an interface and standard base class for all quantum translators in each library. An illustrative example is found in the file cirq_translator.py, where the following code is presented:

class CirqTranslator(QuantumTranslator):      def translate(self, hl_circuit):           import cirq           qubits = [cirq.LineQubit(i) for i in range(hl_circuit.num_qubits)]           circuit = cirq.Circuit()           for gate in hl_circuit.gates:                if gate.name.lower() == "h":                    circuit.append(cirq.H(qubits[gate.qubits[0]]))                elif gate.name.lower() == "cnot":                      circuit.append(cirq.CNOT(qubits[gate.qubits[0]], qubits[gate.qubits[1]]))                elif gate.name.lower() == "x":                     circuit.append(cirq.X(qubits[gate.qubits[0]]))                elif gate.name.lower() == "y":                     circuit.append(cirq.Y(qubits[gate.qubits[0]]))                elif gate.name.lower() == "z":                     circuit.append(cirq.Z(qubits[gate.qubits[0]]))                elif gate.name.lower() == "ry": circuit.append(cirq.ry(gate.params[0])(qubits[gate.qubits[0]]))               # Add other gate translations as needed

Every quantum library translator, like CirqTranslator, QiskitTranslator, and so forth, inherits from the quantum_translator base class. This inheritance structure guarantees a unified interface across all translators, aligning with their respective libraries’ unique syntax and conventions. Within the codebase, these translator classes play a pivotal role by providing essential functionality to seamlessly convert a quantum circuit from its internal representation to the corresponding framework’s representation. This process involves defining a set of shared gates supported by all frameworks capable of executing the translation method with precision, as will be detailed in “Results of the bell state circuit”.

This methodology implies that the predefined rules in Qinterpreter are centered around the established basic gates used in quantum computation. Given that all libraries, with some exceptions, adhere to the basic standard of gates, we embrace this collection of simple quantum gates and adapt them for each specific library, as exemplified in the earlier code in cirq_translator.py. Consequently, the core of the translation process lies in mapping gates from the framework-agnostic model to the corresponding gates in the target framework, ensuring compatibility and successful execution on the chosen backend. Although there are instances where libraries like PennyLane introduce additional functionalities, these are not employed in Qinterpreter due to their deviation from the established standard. Different frameworks may have varying gate sets and representations, requiring careful mapping beyond the scope of this work.

It is crucial to mention that each translator internally uses the backend of its respective library for circuit translation and simulation. For instance, Qiskit uses the Aer backend for state vector simulation, and PennyLane employs the default qubit device for simulation.

Using qinterpreter

Installation and backends

Python was selected as the development platform for the Qinterpreter library due to its straightforward installation process and recognition as a common language supported by leading companies in the quantum industry. This decision is reinforced because the five quantum libraries and their full-stack simulation packages are in Python. Besides, the simplicity of installation is apparent, requiring just a straightforward PIP install command, making it accessible to a diverse user base. All five libraries seamlessly integrate into their respective locations, allowing users to utilize them effortlessly by following the provided instructions in this section.

Depending on the user’s requirements, we offer three installation options to successfully run the Qinterpreter on your local computer. The first method involves cloning the library folder directly from the GitHub repository (Contreras Sepulveda & Contributors, 2023). Once you’ve downloaded the Qinterpreter, you can use the library locally by calling the classes and functions from the same directory.

The second alternative lets users install the Qinterpreter using the “pip install” command directly in the Jupyter console. This can be done by executing the following command in the Python console at the operating system’s shell prompt.

!pip install git+https://github.com/Qubithub/Qinterpreter.git

After the installation process is over, the next step involves importing the requisite libraries. To accomplish this, the users should utilize the following command code.

import math from quantumgateway.quantum_circuit import QuantumCircuit, QuantumGate from quantumgateway.quantum_translator.braket_translator import BraketTranslator from quantumgateway.quantum_translator.cirq_translator import CirqTranslator from quantumgateway.quantum_translator.qiskit_translator import QiskitTranslator from quantumgateway.quantum_translator.pennylane_translator import PennyLaneTranslator from quantumgateway.quantum_translator.pyquil_translator import PyQuilTranslator from quantumgateway.main import translate_to_framework, simulate_circuit

To ensure the proper functionality of the Qinterpreter, the user must consider appropriate versions of the multiple libraries. Table 1 provides a list of these libraries and their corresponding versions.

Table 1 Quantum libraries and their respective versions.

Library	Version	
Qiskit	Qiskit Terra: 0.23.2	
Qiskit Aer: 0.12.0	
Qiskit IBMQ Provider: 0.20.2	
Qiskit: 0.42.0	
Qiskit Nature: 0.6.0	
Pennylane	0.29.1	
Cirq	0.9.1	
Pyquil	3.3.4	
Amazon-Braket	1.36.4	

To cover these requirements, please follow the installation instructions provided below:

1. Qiskit: Install by running the command “pip install qiskit”.

2. Pennylane: Install by running the command “pip install pennylane”.

3. Cirq: Install by running the command “pip install cirq”.

4. Pyquil: Install by running the command “pip install pyquil”.

5. Amazon-Braket: Install by running the command “pip install amazon-braket-sdk”.

Additionally, ensure that your Python and pip versions are up to date. Some packages may require Python 3.6 or later.

The third option involves using our website platform Qubithub.org (https://qubithub.org/), which offers a user-friendly environment for executing Qinterpreter online by visiting the Login page, as shown in Fig. 1.

Figure 1 Displays a screenshot of a user account profile within the Qubithub portal.

Users are introduced to a pre-configured application environment with the necessary libraries already installed, removing the need for manual installation. The user credentials can be obtained by contacting the team. After logging in, the next step involves importing the libraries, as was previously mentioned. This streamlined process allows users to focus more on running their quantum circuits and less on the setup.

Qinterpreter functions

This section presents a detailed overview of the functions employed within the Qinterpreter library. Our primary objective is to provide users with a comprehensive guide and extensive knowledge of the library’s functionalities and capabilities.

Function quantumCircuit()

A circuit is a crucial element in quantum computing, serving as a container for a collection of qubits (Nielsen & Chuang, 2010). Treating these qubits as unified entities allows users to manipulate and modify their states using quantum gates. The QuantumCircuit function, into the Qinterpreter, generates a circuit by carefully considering the number of qubits and classical registers to be incorporated. In this particular scenario, the following code is employed to create a circuit:

circuit = QuantumCircuit(nq,nc)

Here, “nq” represents the number of qubit registers employed, and “nc” denotes the number of classical registers to be defined within the quantum circuit. The defined classical registers are subsequently used for performing measurements.

Function circuit.add_gate()

As previously stated, quantum computing algorithms are commonly depicted using quantum circuit models. These models incorporate quantum gates, projective measurements, and an n-qubit register known as qubits. A vector in the complex space C2describes the state of a qubit. Within this space, quantum gates are represented by unitary complex matrices, reflecting the unitary time evolution of closed quantum systems (Nielsen & Chuang, 2010; Li, Roberts & Yin, 2013; Barenco et al., 1995). In Qinterpreter, we have implemented a comprehensive set of standard gates widely used in the field. The definitions of these gates are presented in Table 2.

Table 2 Matrix representation of the standard used quantum gates.

Gate/Matrix form	Description:	
H=12(111−1)	The Hadamard gate is responsible for setting a qubit into a superposition.	
CNOT=(1000010000001010)	The CNOT gate, also known as the control-not gate, operates on a qubit based on the state of a control qubit, making it a two-qubit gate.	
X=(0110)	The X quantum gate is a gate whose purpose is to flip the state of the qubit along the x-axis over the Bloch sphere. This means that if the qubit is in the |0⟩ state, it will change to the |1⟩ state, and vice versa.	
Y=(0−ii0)	The Y quantum gate serves the same purpose as the X gate, but instead of acting along the x-axis, it operates along the y-axis.	
Z=(100−1)	The Z quantum gate has a similar function to the X and Y gates; however, it operates along the z-axis.	
RX(θ)=(cos⁡(θ2)−isin⁡(θ2)−isin⁡(θ2)cos⁡(θ2))	The RX quantum gate, also known as the rotation X quantum gate, performs a rotation around the x-axis in the Bloch sphere. This rotation is defined by an angle θ, which can be specified as a parameter.	
RY(θ)=(cos⁡(θ2)−sin⁡(θ2)sin⁡(θ2)cos⁡(θ2))	The RY quantum gate, similar to the RX gate, performs a rotation of the qubit along the y-axis at a specified angle. This angle can be defined as a parameter.	
RZ(θ)=(e−iθ200eiθ2)	The RZ quantum gate functions similarly to the RX and RY gates by rotating the qubit along the z-axis.	
CCNOT=(1000000001000000001000000001000000001000000001000000000100000010)	The controlled CNOT gate, also known as CCNOT, is a three-qubit gate that operates on a target qubit based on the states of two control qubits.	
SWAP=(1000001000100001)	The swap quantum gate is a two-qubit gate that interchanges the states of two qubits.	
CP=(100001000010000eiθ)	The controlled Phase quantum gate is a two-qubit gate that modifies the phase angle of a target qubit based on the state of a control qubit.	
CZ=(100001000010000−1)	The controlled Z quantum gate is a two-qubit gate that performs a controlled-Z operation, which means that it applies a Z (Pauli-Z) gate to the target qubit if and only if the control qubit is in the state |1⟩.	
CY=(10000100000−i00i0)	The controlled Y quantum gate is a dual-qubit gate designed to execute a controlled-Y operation; this implies that it applies a Y (Pauli-Y) gate to the target qubit only when the control qubit is in the |1⟩ state. If the control qubit is in the state |0⟩, the target qubit remains its original state without any modification.	
Measure	The measurement is not technically considered a gate in the same sense as other quantum gates, but it is an operation that acts on a qubit, causing it to collapse into one of the possible measurement outcomes.	

In this instance, we present the gates currently implemented in Qinterpreter. Table 3 illustrates the procedure for incorporating these gates into any circuit object.

Table 3 Source code of the set standard quantum gates defined in Qinterpreter.

Gate	Code	
H	circuit.add_gate(QuantumGate ("h", [0]))	
CNOT	circuit.add_gate(QuantumGate ("cnot", [0, 1]))
//Contro: q0, Objective: q1	
X	circuit.add_gate(QuantumGate("x", [0]))	
Y	circuit.add_gate(QuantumGate("y", [0]))	
Z	circuit.add_gate(QuantumGate("z", [0]))	
RY	circuit.add_gate(QuantumGate("ry", [0], [Angle]))
//Rotate Y axis by any angle	
RX	circuit.add_gate(QuantumGate("rx", [0], [math.pi/2])) //Rotate X axis by any angle	
RZ	circuit.add_gate(QuantumGate("rz", [0], [math.pi/2])) //Rotate Z axis by any angle	
CCNOT	circuit.add_gate(QuantumGate("toffoli", [0,1,2]))
//Control: q0 and q1, Objective: q2	
SWAP	circuit.add_gate(QuantumGate("x", [0, 1]))
//Swap between q0 and q1	
CP	circuit.add_gate(QuantumGate("CPhase", [0, 1], [Angle]))
// Applied an angle	
CZ	circuit.add_gate(QuantumGate("cz", [0, 1]))
//Contro: q0, Objective: q1	
CY	circuit.add_gate(QuantumGate("cy", [0, 1]))
//Contro: q0, Objective: q1	
Measure	circuit.add_gate(QuantumGate("MEASURE", [i, j]))
//Where i is the qubit register index and j is the classical register index.	

Note: The Toffoli gate is a standard quantum computing gate that modifies the state of a target qubit based on two control qubit states. In addition, Toffoli gates can be achieved through a sequence of elementary quantum gates (Li, Roberts & Yin, 2013; Barenco et al., 1995; Roe, 1998).

Function translate_to_framework()

The Qinterpreter library serves the purpose of translating instructions to various quantum computing frameworks. The Qinterpreter library is currently compatible with five libraries: Qiskit, Pyquil, Cirq, Pennylane, and Amazon-Braket. These libraries were selected based on their GitHub metrics, which can be interpreted as a measure of popularity. To choose the desired framework (Qiskit, Pyquil, Pennylane, Amazon-Braket, or Cirq), for executing on our circuit, the following code is used:

selected_framework = ′qiskit′ translated_circuit = translate_to_framework(circuit, selected_framework)

Here, the variable “selected_framework” can take one of the following values: {qiskit, cirq, pennylane, pyquil, amazonbraket}.

Function translated_circuit.print_circuit()

Printing the circuit in any quantum computing library allows users to visualize and debug their created quantum circuit. This functionality is also implemented in the Qinterpreter framework through the “print_circuit” function. In short, once the framework has been selected, we can print our previously defined circuit (as described in the subsection “Defining a QuantumCircuit”) by specifying the following code:

translated_circuit.print_circuit()

Function simulate_circuit()

We use the appropriate simulators provided by each library to simulate a specific quantum circuit. For example, in the case of Qiskit, we utilize the QASM simulator. However, when using the Pyquil framework, the user must install the necessary software requirements. This information can be found in the “Installation and Getting Started” section of the pyQuil documentation on the Rigetti website. The documentation provides instructions on installing the required software and executing the commands. The command to perform and print the simulations is as follows:

print(simulate_circuit(circuit, selected_framework))

Please note that the simulation will only be executed if one in the circuit performs the measurement function. These measurement functions should be applied before running the simulate_circuit(). To apply the measurement function, the user needs to run the following code:

circuit.add_gate(QuantumGate("MEASURE", [i,j]))

In the code snippet, [i,j] represents the indices of the qubit register and classical register where the quantum measurement function will be applied.

There is no need to modify the interpreter to execute any algorithm. Users can write their algorithm following the rules of Qinterpreter and specify their desired framework (Qiskit, Pyquil, Cirq, Pennylane, Braket) for code execution. The interpreter will seamlessly convert the Qinterpreter instructions into the appropriate instructions for the selected framework. The execution will use the resources provided by the chosen framework and the resulting outcomes will be presented to the user. The following section will illustrate this process through three main examples.

Applications of the qinterpreter

To show the use of the Qinterpreter, we reproduce three well-known examples in the field of quantum computing. These examples are: Bell States algorithm.

Grover algorithm

Shor Algorithm for the numbers 15, 21 and 35.

The first example is a straightforward demonstration of creating Bell states, which are fundamental entangled states in quantum computing. The second example refers to the well-known Grover algorithm, and the third example involves the principles of the Shor algorithm to solve a specific problem related to factoring in the numbers 15, 21, and 35. Both algorithms are implemented using the previously mentioned frameworks: Qiskit, Pyquil, Cirq, Pennylane, and Braket.

Bell’s states

In this first example, we simulate a basic quantum circuit that aims to generate one of the four Bell’s states. The Bell states are those maximally entangled, and arise from a superposition of the quantum bits’ basis states |0⟩ and |1⟩, defined by

(1) |ψ+⟩=12(|00⟩+|11⟩),

(2) |ψ−⟩=12(|00⟩−|11⟩),

(3) |φ+⟩=12(|01⟩+|10⟩),

(4) |φ−⟩=12(|01⟩−|10⟩).

These states can be emulated using quantum computers (Audretsch, 2007; Ou, 2007; Scully & Zubairy, 1997). In this case, we reproduce the most common one, “Bell state 00”, defined by the Eq. (1). To achieve this, we will use two specific gates: the Hadamard gate (H), used to put a qubit in a superposition state, and the Controlled-NOT (CNOT), a two-qubit gate that flips the state of a qubit based on the value of a control qubit. It is important to note that initially, it is necessary to follow the instructions in “Bell’s States”, which involve cloning the Qinterpreter repository and importing the required libraries. Once this is done, we will create a circuit with two qubits and two classical bits, as shown below:

n = 2 circuit = QuantumCircuit(n,n)

In this case, we import two quantum registers and two classical registers, as the creation of the first Bell state requires two quantum bits and two classical registers for the simulation. We can then add the gates described earlier to our circuit.

circuit.add_gate(QuantumGate("h", [0])) circuit.add_gate(QuantumGate("cnot", [0, 1]))

where the two specific gates: the H gate to the first qubit (0), creates a superposition, making the state 12(|00⟩+|01⟩), whereas the CNOT gate, with the first qubit, 0, as the control and the second qubit, 1, as the target. This entangles the qubits and creates the Bell state |ψ+⟩.

To perform the simulation of our circuit, we implemented the measurement operation as follows:

circuit.add_gate(QuantumGate("MEASURE", [0, 0])) circuit.add_gate(QuantumGate("MEASURE", [1, 1]))

Here, the measurement gates are added to the circuit for both qubits, indicating that the quantum state will be measured. Next, we choose the framework for displaying our simulation, and in our case, for the sake of simplicity, we initially opt for IBM’s Qiskit.

selected_framework = ′qiskit′ # Change this to the desired framework translated_circuit = translate_to_framework(circuit, selected_framework)

To visualize the circuit and ensure it works correctly, we use the “print_circuit()” to print the circuit

translated_circuit.print_circuit()

Finally, we simulate the circuit using the following command:

print("The results of our simulated circuit are: ") counts=simulate_circuit(circuit, selected_framework) from qiskit.visualization import plot_histogram plot_histogram(counts, title ="Histogram of Quantum States")

Results of the bell state circuit

We are going now analyze the results from our previous circuit, as presented in Table 4. In each column of the table, we find the framework name used, the circuit obtained through printing on each framework, and the outcomes of the simulations conducted within each framework.

Table 4 Presents the results of "Bell state 00" within each of the five different frameworks.

Framework	Framework graph	Simulation	
Qiskit		The results of our simulated circuit are: {‘00’: 504, ‘11’: 520}	
Cirq		The results of our simulated circuit are: {‘11’: 523, ‘00’: 477}	
Pennylane		The results of our simulated circuit are:
{‘00’: 521, ‘11’: 479}	
Amazon-Braket		The results of our simulated circuit are: {‘00’: 490, ‘11’: 510}	
Pyquil		The results of our simulated circuit are: Counter ({‘00’: 51, ‘11’: 49})	

The first part: ‘00’: 504 means: |00⟩→Indicates that |qubit0⟩ is in the state |0⟩, and the |qubit1⟩ is in the state |0⟩ and the simulation process yielded this result 504 times. In the second segment, labeled ‘11’:520 means: |11⟩→Indicates that |qubit0⟩ is in the state |1⟩ and |qubit1⟩ is in the state |1⟩, and the simulation process found that result 520 times. The results are visualized through the following histogram graph in Fig. 2.

Figure 2 Illustrates the measurement outputs for the Bell state 00 executed in Qiskit.

A noteworthy observation concerning the preceding code is that it can be easily adapted to generate the “Bell State 01”, denoted as |ψ−⟩, rather than the 00-state represented by |ψ+⟩. This adjustment involves a minor modification in the gate sequence. Specifically, after applying the Hadamard gate to the first qubit, an X gate needs to be used to flip the state of the first qubit, resulting in the desired Bell State 01. The modified part of the code is provided below:

circuit.add_gate(QuantumGate("h", [0])) circuit.add_gate(QuantumGate("x", [0])) circuit.add_gate(QuantumGate("cnot", [0, 1]))

with these adjustments, and while keeping the remainder of the code unchanged, including the measurement and simulation steps, the code effectively gives the simulation of the desired entangled state. Additionally, to generate the “Bell State 10”, represented by |φ+⟩, an X gate must be applied to the second qubit after the Hadamard gate is applied to the first qubit. Finally, for the “Bell State 11”, denoted by |φ−⟩, an X gate must be applied to both qubits after the Hadamard gate is applied to the first qubit.

On the other hand, it is crucial to ensure the proper library for visualization is imported when examining the histogram section that illustrates the outcomes of each framework. These commands are explained in detail within the preloaded file “Bell States.ipynb,” available on the GitHub Qinterpreter page upon download (Contreras Sepulveda & Contributors, 2023). For a visual understanding of each qubit’s quantum state and operations, we have developed an automated function within Qinterpreter. In this instance, we employ the Bloch Sphere Visualization Function (bloch_sphere(circuit)). This function sequentially translates the circuit, conducts simulation, calculates the reduced density matrix, and visualizes qubit states on Bloch spheres (Ömer, 2009; IBM Quantum, 2023; Lu, Mia & Metcalf, 2005). This function emulates and extends similar functionalities offered by the Qiskit library. Consequently, it generates a Plotly interactive figure showcasing individual Bloch spheres for each qubit and their corresponding state vectors and markers. An illustrative example is provided below:

from quantumgateway.main import translate_to_framework, simulate_circuit, bloch_sphere import math circuit = QuantumCircuit(2,2) circuit.add_gate(QuantumGate("x", [0])) circuit.add_gate(QuantumGate("x", [1])) circuit.add_gate(QuantumGate("h", [1])) circuit.add_gate(QuantumGate("cphase", [0, 1], [math.pi/4])) bloch_sphere(circuit)

such an example involves a two-qubit circuit, where two consecutive X gates are applied to the first and second qubits. Following this, an H gate is applied to the second qubit, and a controlled-phase (CPhase) gate is applied between the first and second qubits, featuring a rotation angle of π/4. The resulting interactive Bloch sphere visualization is displayed upon executing the bloch_sphere function with the given circuit. The visualization depicts two spheres, each corresponding to one of the qubits in the circuit, as observed in Fig. 3. It effectively showcases the final state of the qubits in their respective spheres. It is essential to mention that one must install Plotly to use this function, which can be easily done with the command !pip install plotly.

Figure 3 Bloch sphere representation for each qubit.

Grover’s algorithm

In the second example, we explore a quantum search algorithm to locate an element within an unorganized list of elements. Grover’s algorithm (Grover, 1996; Perkowski, 2022) is renowned in quantum computing for its efficiency, surpassing classical search algorithms like the binary search algorithm (Knuth, 1998). Grover’s algorithm consists of three crucial steps: 1. First step: The initialization:

In this initial stage, our system is prepared to implement Grover’s algorithm. This involves bringing all the qubits into a superposition state by applying a Hadamard gate to each qubit. Consequently, the state of our system becomes:

(5) |s⟩=1N∑x=0N−1|x⟩,

where N is the total number of states. 2. Second step: Oracle application

Following the initialization, we apply an oracle to the element we intend to search. This is accomplished using a reflection operator Uω on the target element. For example, if |w⟩ represents the element we are looking for, the oracle function reflects the state in a manner that designates it as the target:

(6) Uω|x⟩={|x⟩ifx≠ω−|x⟩ifx=ω

Hence, if the searched element is |w⟩, its phase will be inverted. 3. Third step: Amplitude amplification

In the final step, we modify our quantum space by reflecting around the average amplitude |w⟩. This is achieved by applying the operator Us=2|s⟩⟨s|−I. Consequently, the amplitudes of all qubits are adjusted, except the state |w⟩, which undergoes amplification.

Now, let’s explore an example for a two-qubit system in our Qinterpreter library. We begin by importing the necessary libraries and modules and after then we start from the initial state |ψ⟩=|00⟩, and apply H gates to each qubit to create a superposition state

(7) |ψ⟩=H|00⟩=12(|00⟩+|01⟩+|10⟩+|11⟩),

in code, this initialization is implemented as:

circuit = QuantumCircuit(2, 2) for i in range(2):      circuit.add_gate(QuantumGate("h", [i]))

Once our Hadamard gates are ready, we can apply an Oracle targeting the state |11⟩. The apply_oracle function adds gates to the circuit to create an Oracle focusing on the state |11⟩.

def apply_oracle(circuit):      circuit.add_gate(QuantumGate("x", [0])) # Change qubit 0 to |1⟩      circuit.add_gate(QuantumGate("x", [1])) # Change qubit 1 to |0⟩      circuit.add_gate(QuantumGate("cphase", [0, 1], [-math.pi])) # Apply cphase with negative phase      circuit.add_gate(QuantumGate("x", [0])) # Restore qubit 0 to |0⟩      circuit.add_gate(QuantumGate("x", [1])) # Restore qubit 1 to |0⟩

In this case, we use an oracle for the state |11⟩ by using the cphase gate with an angle −π as our oracle matrix Uw:

(8) Uw=(100001000010000−1),

When applied to our system, we obtain

(9) |ψ⟩=Uw|ψ⟩=12(|00⟩+|01⟩+|10⟩−|11⟩),

Subsequently, we apply the diffusion operator Ud=2|s⟩⟨s|−I, to increase the probability of our target state. The operation sequence is as follows: After Oracle application:

(10) |ψ⟩=12[111−1],

Applying the diffusion operator:

(11) |ψ⟩′=D|ψ⟩=[00.50.50.50.500.50.50.50.500.50.50.50.50][1/21/21/2−1/2]=[0.250.250.250.75],

As we observe, the |11⟩ state has the highest probability.

In code, the diffusion operator is implemented as follows:

# Function to add the Amplification Operator def apply_amplification(circuit):       # Apply Hadamard and X gates      for i in range(2):           circuit.add_gate(QuantumGate("x", [i])) # Apply X gates before Hadamard           circuit.add_gate(QuantumGate("h", [i])) # Sequence of gates for the Amplification Operator      pi = math.pi      circuit.add_gate(QuantumGate("cphase", [0, 1], [-pi])) # Adjustss the phase of the cphase # Apply Hadamard and X gates again      for i in range(2):           circuit.add_gate(QuantumGate("h", [i]))           circuit.add_gate(QuantumGate("x", [i])) # The previously defined functions apply_oracle and apply_amplification are called to apply # the Oracle and the Amplification Operator to the circuit apply_oracle(circuit) apply_amplification(circuit)

We incorporate measurement gates and select the desired framework for executing our Grover algorithm. In Qinterpreter, this is accomplished through

#----------------------------------------------------------------- # Measurement of Qubits #----------------------------------------------------------------- for i in range(2): circuit.add_gate(QuantumGate("measure", [i, i])) # Measurement gates are added to measure the state of each qubit. #----------------------------------------------------------------- # Step 8) Simulating the Circuit and Printing the Result #----------------------------------------------------------------- # we can proceed to see what the quantum circuit looks like in Xanadu's PennyLane selected_framework = ′pennylane′ # Change this to the desired framework translated_circuit = translate_to_framework(circuit, selected_framework) translated_circuit.print_circuit() # and the visualization of the results are print("The results of our simulated circuit are: ") print(counts) import matplotlib.pyplot as plt states = [′00′, ′01′, ′10′, ′11′] counts_keys = list(counts.keys()) result_counts = {state: counts[state] if state in counts_keys else 0 for state in states} total_counts = sum(result_counts.values()) percentages = {state: (count / total_counts) * 100 for state, count in result_counts.items()} plt.bar(percentages.keys(), percentages.values()) plt.xlabel(′State′) plt.ylabel(′Measurement Probability (%)′) plt.title(′Grover Algorithm Simulation Results (Pennylane)′) plt.ylim(0, 110) plt.show()

Finally, we integrate measurement gates and choose the desired framework for executing our Grover algorithm in Qinterpreter. In our case, we have opted for the Pennylane framework. Figure 4 depicts the quantum circuit diagram for a 2-qubit system in this framework, and the measurement probability for the 2-qubit state is illustrated in Fig. 5, respectively. Our 2-qubit system circuit exhibits no errors; thus, it is free from random noise in the system.

Figure 4 Quantum circuit diagram for the state |11⟩ in the Pennylane framework.

Figure 5 Measurement probability of 2-qubit system for the state 11.

Furthermore, with minor modifications to the code, one can easily adapt the algorithm to find other respective states. For instance, adjusting the Oracle part of the circuit to match the state |10⟩ instead of |11⟩ involves a simple modification, specifically:

def apply_oracle(circuit):      circuit.add_gate(QuantumGate("x", [1])) # Change qubit 1 to |0⟩      circuit.add_gate(QuantumGate("cphase", [0, 1], [-math.pi]))      circuit.add_gate(QuantumGate("x", [1])) # Restore qubit 1 to |0⟩

Similarly, if the goal is to modify the Grover algorithm to find the state |01⟩ instead of |10⟩, starting from the code designed to find |11⟩, a minor correction in the Oracle part of the circuit is necessary, as follows:

def apply_oracle(circuit):      circuit.add_gate(QuantumGate("x", [0]))      circuit.add_gate(QuantumGate("cphase", [0, 1], [-math.pi]))      circuit.add_gate(QuantumGate("x", [0]))

and to find the state |00⟩, starting from the code designed to find |11⟩, we need to make changes to both the Oracle and Amplification Operator. In the Oracle part, it is necessary to remove all x gates and leave only the cphase gate, whereas in the amplificatory operator part, we must change the angle of the cphase gate. Instead of -pi, it will be pi. We have also developed Grover’s quantum search algorithm for 4-qubit systems, showcasing an absence of noise and errors in the system design (Jingle et al., 2022). Comprehensive algorithm explanations are available in the Qinterpreter files (Contreras Sepulveda & Contributors, 2023).

Shor’s factorization algorithm

In this subsection, we implement the Shor Algorithm by using the Qinterpreter. The Shor Algorithm is a quantum computing algorithm that enables the identification of prime factors for any given integer (see Shor, 1999, 1994; LaPierre, 2021; Yimsiriwattana & Lomonaco, 2004; Ekert & Jozsa, 1996). To employ the Qinterpreter in developing the Shor Algorithm and simulate it across various frameworks such as Qiskit, Cirq, Pyquil, Pennylane, and Amazon-Braket, we proceed by creating our circuit and incorporating the measurement gates. To use the Qinterpreter and implement the Shor Algorithm, we import the necessary modules and create our circuit, incorporating the required measurement gates

from math import pi from fractions import Fraction import math import numpy as np #----------------------------------------------------------------- #Assigning Constants: #----------------------------------------------------------------- pi = math.pi # П is assigned to the variable ′pi′ n_count = 4 #----------------------------------------------------------------- #Quantum Circuit Initialization: #----------------------------------------------------------------- circ = QuantumCircuit(8, 8) #----------------------------------------------------------------- #Applying initial Quantum Gates: #----------------------------------------------------------------- for i in range[4]: circ.add_gate(QuantumGate("H", [i])) # An Pauli-X gate is applied to the 5th qubit circ.add_gate(QuantumGate("X", [4])) # CNOT gates are applied to different pairs of qubits circ.add_gate(QuantumGate("CNOT", [0, 5])) circ.add_gate(QuantumGate("CNOT", [0, 6])) circ.add_gate(QuantumGate("CNOT", [1, 4])) circ.add_gate(QuantumGate("CNOT", [1, 6] )) # Toffoli gates are applied to control qubits 0 and 1, targeting qubits 4, 5, 6, and 7. for i in range(4, 8):      circ.add_gate(QuantumGate("Toffoli", [0, 1, i])) #----------------------------------------------------------------- #Measure the auxiliary qubits: #----------------------------------------------------------------- # MEASURE gates are applied to the auxiliary qubits (qubits 4, 5, 6, and 7). for i in range(4, 8):      circ.add_gate(QuantumGate("MEASURE", [i, i])) #----------------------------------------------------------------- #Quantum Fourier Transform (QFT): #----------------------------------------------------------------- n=4 for i in range(n-1, -1, -1):      circ.add_gate(QuantumGate("H", [i]))      for j in range(i - 1, -1, -1):           circ.add_gate(QuantumGate("CPHASE", [j, i], [pi/(2 ** (i - j))])) #Controlled-phase gates (CPHASE) are then applied to implement the Quantum Fourier Transform for i in range(n // 2):      circ.add_gate(QuantumGate("SWAP", [i, n - i - 1])) #SWAP gates are used to reorder qubits in the QFT, and MEASURE gates are applied to the control qubits #----------------------------------------------------------------- #Measure the control qubits: #----------------------------------------------------------------- for i in range(4):      circ.add_gate(QuantumGate("MEASURE", [i, i+4]))

Selecting the desired framework:

selected_framework = ′qiskit′ # Change this to the desired framework translated_circuit = translate_to_framework(circ, selected_framework)

Printing the circuit to visualize the results:

translated_circuit.print_circuit()

As the circuit is rather extensive, for the sake of simplicity, we present only the results from the Qiskit framework in Fig. 6. However, users have the option to simulate the circuit using the alternative frameworks.

Figure 6 Schematizes the quantum circuit for the compiled version of Shor’s algorithm in Qiskit.

To print our results, we use the following command code

print("The results of our simulated circuit are: ") counts = simulate_circuit(circ, selected_framework) print(counts) #----------------------------------------------------------------- #Analyze Measurement Results: #----------------------------------------------------------------- # Convert binary to decimal and remove zeros measured_values = {int(k[:n_count], 2) for k in counts.keys() if int(k[:n_count], 2) != 0} print("Measured values:", measured_values)

The results of our simulated circuit are:

{′11000000′: 224, ′10000000′: 255, ′01000000′: 252, ′00000000′: 269}

These results correspond to the binary numbers: 12, 8, 4, and 0, where the number 12 was obtained 224 times, the number 8 was obtained 255 times, the number 4 was 252 times and the 0 number was 269.

Interpreting these results according to the Shor algorithm procedure: Try to find the period ‘r’ and the factors of ‘N’ for each ‘a’ where a is an aleatory number between 2 and 15 different from a factor of 15. This is done through the following steps:

Initializes an empty list to store estimates for period ‘r’ (estimates = []).

Iterates over the values measured by the quantum circuit (form m in measured_values:). Each measured value is an estimate of period ‘r’.

Calculates an estimate of ‘r’ as the denominator of the fraction m/2^n (estimate = Fraction (m, 2**n_count) and r = estimate.denominator). Python’s Fraction class reduces the fraction to its simplest form, and the denominator of this simplified fraction is an estimate of the period ‘r’.

Checks if a^r mod 15 is equal to 1 (if pow(a, r, 15) == 1:). If so, this ‘r’ is a good estimate of the period and can be used to find the factors of ‘N’.

Computes two candidate factors of ‘N’ as a^(r/2) ± 1 and finds their common factors with ‘N’ (factor1 = math.gcd (a**(r//2) + 1, 15) and factor2 = math.gcd (a**(r//2) − 1, 15)). If any of these common factors is greater than 1 and has not been found before, it is added to the set of factors.

If a period is not found for a value of ‘a’, a message is displayed to the console (print ("Did not find a period.")).

The Shor algorithm generates the factors of the number 15: {3, 5}. The corresponding code for obtaining these results is provided below:

#----------------------------------------------------------------- #Factor Finding with Shor's Algorithm: #----------------------------------------------------------------- factors = set() prime_factors = set() # Initializes two sets, factors and prime_factors, to store the factors and prime factors obtained during the algorithm for a in range(2, 15):      if math.gcd(a, 15) != 1:           continue      found_period = False      for m in measured_values:           r = Fraction(m, 2 ** n_count).denominator           if pow(a, r, 15) == 1 and r % 2 == 0: # For each valid a, it searches for a period r such that a^r(mod15)≡1 and r is even.                factor1 = math.gcd(a ** (r // 2) + 1, 15)                factor2 = math.gcd(a ** (r // 2) - 1, 15)                factors.update([factor1, factor2]) # If a valid period is found, it calculates potential factors using the period and updates the factors set.                    found_period = True          if not found_period:                print(f"Did not find a period for a = {a}.") # Check if the factors are prime for factor in factors:      if factor > 1 and all(factor % i != 0 for i in range(2, int(math.sqrt(factor)) + 1)): #     Checks if the factors in the factors set are prime.           prime_factors.add(factor) #     Adds prime factors to the prime_factors set. line = "*" * 70 # Prints a line of asterisks for visual separation. print(line) if prime_factors:      print("The prime factors of the number 15, using the Shor Algorithm are:", prime_factors) # If there are prime factors, it prints them. else:      print("No prime factors found.") print(line)

Similar to the Grover algorithm’s code, we can make minor adjustments to the preceding code to find the prime factors of the numbers 21 and 35. For 21, the prime factors are 3 and 7, and for 35, they are 7 and 5. Of course, these adjustments involve modifying the number of qubits. Specifically, the factorization of 21 requires a total of five qubits: three for the control register and two for the work register. In contrast, the factorization of 35 requires a total of eight qubits. Moreover, some adjustments in the loop ranges and changes in the iteration range for ‘a’ must also be made. For the sake of simplicity, we have only included the corresponding quantum circuit diagrams for the factorization of 21 and 35 in Figs. 7 and 8, respectively, using the Amazon Braket framework.

Figure 7 Full circuit diagram for the Shor’s factoring algorithm for number 21.

Figure 8 Full circuit diagram for the Shor’s factoring algorithm for number 35.

We have thoroughly outlined the strengths and advantages of Qinterpreter. Currently, Qinterpreter has not yet been tested for executing circuits and objectives on real quantum devices through their respective interfaces due to a lack of access to cloud quantum computing services, as offered by quantum computing environments like Qiskit. For example, if we get access to real quantum devices, the Qinterpreter needs to be extended to capture the full spectrum of noise and error characteristics inherent in such devices. This disparity between simulated and actual quantum behavior can impact the accuracy of predictions.

Moreover, executing complex quantum algorithms and simulations may demand significant computational power and memory. This poses limitations on the size and scope of simulations that can be effectively performed on current quantum computers, making it less than ideal for individual researchers needing a dedicated and faster simulator for their experiments. Simulating systems with a large number of qubits becomes computationally demanding, with resources growing exponentially as the number of qubits increases.

Thus, there are restrictions on the number of qubits in each quantum backend of the Qinterpreter simulator. For instance, IBM’s simulator has a maximum size of 32 qubits, and expansion is not an option. AWS Braket, on the other hand, can accommodate up to 34 qubits. The base simulator in Cirq can calculate circuit results for up to 20 qubits using cirq.Simulator(). The Pennylane simulator can handle circuits with up to 28 qubits. Finally, the PyQuil simulator, known as the QVM (Quantum Virtual Machine), has a limitation of up to 26 qubits.

Despite lacking access to real quantum devices, the Qinterpreter simulator provides students and newcomers with exposure to various quantum computing environments. This knowledge proves essential for researchers and practitioners, aiding them in choosing the most suitable simulator aligned with their unique quantum computing needs. As a result, navigating these interfaces enhances the learning experience by introducing students to various environments, elucidating simulator limitations, emphasizing considerations in resource management, and shedding light on scalability challenges. These advantages collectively contribute to a well-rounded and practical education for newcomers in the field of quantum computing.

Conclusions and future work

We have introduced a quantum interpreter that plays a significant role in combining the five most popular Python-based quantum libraries into a unified framework. It is offered via a science gateway that can be installed locally or used in a Python environment. Through the replication of three well-known quantum computing examples, we have effectively demonstrated the Qinterpreter feasibility, providing the user with a generic and seamless experience similar to that of a classical interpreter. Additionally, the Qinterpreter incorporates other well-known algorithms, such as the HHL algorithm for solving linear equation systems (Harrow, Hassidim & Lloyd, 2009). This particular algorithm, initially crafted by former developers (Benson, 2023), has been subsequently revised by us to align with the latest Quiskit libraries, accompanied by detailed explanations of their implementations. We also envision the potential extension of Qinterpreter’s source code to support other applications where there exists an incentive to explore and broaden Qinterpreter’s capabilities to support additional programming languages, such as Julia, fostering collaboration among diverse groups. This initiative aims to enhance engagement and improve the accessibility of quantum computing education.

Therefore, we firmly believe that Qinterpreter has the potential to make a significant impact in the field of quantum computing. Looking ahead, we also envision a Qinterpreter role in quantum machine learning (QML). Future endeavors will focus on implementing a wide range of QML algorithms on different platforms and exploring practical applications in various domains. For instance, QML could prove beneficial in computationally demanding tasks like density functional theory calculations for solving many-body wavefunctions (Gaitan & Nori, 2009; Senjean, Yalouz & Saubanère, 2023) or quantum spectral clustering (Kerenidis & Landman, 2021). Additionally, the trainability of QML models opens up possibilities for modeling larger DNA molecules, like the G-quadruplex (Phan et al., 2015; de Luzuriaga et al., 2022; Khoshbin et al., 2020). In the long term, we believe that our efforts will bring us closer to creating an accessible and user-friendly quantum computing environment on our Qubithub platform, benefitting the Mexican community and other Latin American and international communities. In this arena, the goal is to contribute novel educative and training content as an alternative or complementary education in a science gateway portal, promoting diversity, inclusion, and fostering interest in quantum computing within Hispanic regions and beyond.

Additional Information and Declarations

Competing Interests

Author Contributions

Data Availability

The authors declare that they have no competing interests. Sandra Gesing is an Academic Editor for PeerJ Computer Science.

Wilmer Contreras-Sepúlveda conceived and designed the experiments, performed the experiments, analyzed the data, performed the computation work, prepared figures and/or tables, authored or reviewed drafts of the article, and approved the final draft.

Braulio Misael Villegas-Martínez conceived and designed the experiments, performed the experiments, analyzed the data, performed the computation work, prepared figures and/or tables, authored or reviewed drafts of the article, and approved the final draft.

Sandra Gesing conceived and designed the experiments, performed the experiments, analyzed the data, prepared figures and/or tables, authored or reviewed drafts of the article, and approved the final draft.

José Javier Sánchez-Mondragón performed the experiments, analyzed the data, prepared figures and/or tables, authored or reviewed drafts of the article, and approved the final draft.

Juan Carlos Sánchez-Pérez conceived and designed the experiments, performed the experiments, analyzed the data, performed the computation work, prepared figures and/or tables, authored or reviewed drafts of the article, and approved the final draft.

Claudia Andrea Vidales-Basurto conceived and designed the experiments, performed the experiments, analyzed the data, performed the computation work, prepared figures and/or tables, authored or reviewed drafts of the article, and approved the final draft.

J. Jesús Escobedo-Alatorre analyzed the data, prepared figures and/or tables, authored or reviewed drafts of the article, and approved the final draft.

Angel David Torres-Palencia analyzed the data, prepared figures and/or tables, authored or reviewed drafts of the article, and approved the final draft.

Omar Palillero-Sandoval analyzed the data, prepared figures and/or tables, authored or reviewed drafts of the article, and approved the final draft.

Jacob Licea-Rodriguez analyzed the data, prepared figures and/or tables, authored or reviewed drafts of the article, and approved the final draft.

Néstor Lozano-Crisóstomo analyzed the data, prepared figures and/or tables, authored or reviewed drafts of the article, and approved the final draft.

Julio César García-Melgarejo analyzed the data, prepared figures and/or tables, authored or reviewed drafts of the article, and approved the final draft.

Eddie Nelson Palacios-Perez analyzed the data, prepared figures and/or tables, authored or reviewed drafts of the article, and approved the final draft.

The following information was supplied regarding data availability:

The code is available at Zenodo: Contreras Sepúlveda, W., Villegas Martínez, B. M., & Sánchez Pérez, J. C. (2024). Qinterpreter (1.0). Zenodo. https://doi.org/10.5281/zenodo.10652483.

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
