# Peer review of "Unleashing quantum algorithms with Qinterpreter: bridging the gap between theory and practice across leading quantum computing platforms"

_PeerJ Computer Science, doi:10.7717/peerj-cs.2318_

## Round 0.1 · original submission · Major Revisions

The manuscript is reviewed by two independent reviewers who are experts in this field. They observed that the manuscript needs revisions in terms of the presentation and the experimental analysis. Just a description of the library alone would not be of interest to the readers. Hence authors are suggested to pay close attention to the comments given by the reviewers and revise the manuscript.

Reviewer 1 ·

Basic reporting

In paper Authors provided a tool to unifying process of quantum circuits creation by the use of other widely known software solutions like e.g. Qiskit or PennyLane. In general such tool will be valuable for general quantum community and very usable for beginner user of quantum systems but also for senior researches which want to perform some experiment with one tool for different backends. Hide other technical details with one API or interface may be naturally valuable for other researchers which are interested directly with quantum experiment but there are not interested with other computer/programming details.

Experimental design

In paper Authors give some general but still relevant arguments for their solutions. However, in my opinion some part of paper still requires update. Authors give some general information about installation, give arguments for popularity of their software and its role in Hispanic community which naturally are very important arguments and remarks. But, presentation of Shor algorithm could be better, e.g. other numbers for factorization should be also presented, e.g. 25, 35 and etc. Also other quantum algorithm like Grover’s algorithm should be presented. Authors should provide examples for other known today algorithms e.g. solution of linear equations system or/and example of circuit from quantum machine learning e.g. quantum spectral clustering.

Validity of the findings

Presentation of qinterpreter also should be improved. Authors omitted some other important problems: e.g. how we can perform measurement operation e.g. in different than standard base. How we can design an user unitary gate or an user designed controlled gate i.e. how to define other unitary gates from universal set of gates. The algorithm of translation from qinterpreter to e.g. Qiskit backend is not discussed. Authors should provide more details how the translation from qinterpreter quantum circuits to other backend is performed. Some times visualization of quantum on the Bloch Sphere is also very valuable but paper does not contains any discussion on thus problem.

Additional comments

Therefore, in my opinion Authors should extend paper with several more advanced examples of quantum circuits and also supplied some examples with definitions of new user unitary gates and extend paper with examples with different measurement operations. So, the major revision is required for current version of paper. Examples given in paper we have only two very known examples of quantum circuits so examples of other circuits should significantly improves the final score of paper.

·

Basic reporting

The author tries to explain the use of qinterpreter in Q world. I suggest the following changes/additions to this article.
- Motivation/Introduction: The author is too much engrained in explaining Quantum computing and classical computing. Instead, it needs to focus on the prime purpose of the article, which is the Qinterpreter. Replace it with an explanation of the Qinterpreter and currently available features.
- Why should someone use this library?
- Limitation: What are limitations? Limitations include Qubits, platforms, etc.
- Which platform is it available?

Experimental design

No Design in the article. Should be treated as a essay.

Validity of the findings

No Findings in the article. More of explanation of library.

Additional comments

No comments

---

## Round 0.2 · Major Revisions

The manuscript is interesting and on an emerging topic. However, reviewers comment on the rigour, technical and presentation aspects. Authors must revise the manuscript and provide a detailed revision comments.

Reviewer 1 ·

Basic reporting

Authors improved several aspects of the reviewed paper. But, in my opinion presentation of pieces of source code should be changed in the final version. Currently black background is not a good solution from clarity point of view. Especially, if someone would like to print the article. But, other parts of the paper technically sound good.

Experimental design

no comment

Validity of the findings

no comment

Additional comments

Authors given additional remarks into reviewed paper and they answered on comments given at the first review. In my opinion introduced changes improved the final score of paper, therefore the reviewed paper may be accepted for publication after naturally additional reread and final checks of technical quality.

Reviewer 3 ·

Basic reporting

The reporting is clear with professional english. But there are spelling mistake at couple of places noticed.
The literature references are also sufficiently good. No problem
The structure, figures and tables are Ok.
Authors provided a tool to unifying process of quantum circuits creation by the use of widely known software solutions like e.g. IBM Qiskit, Amazon Braket, Cirq, PyQuil, and PennyLane. In general such tool will be valuable for general quantum community and very usable for beginner user of quantum systems but also for senior researches which want to perform some experiment with one tool for different backends. This is more of a help kind of document for the tool which is developed to unified application approach. This tools help to use quantum simulator only and not specific to any particular quantum real hardware. The scope is well stated in the proposal. This tool helps the beginners of quantum computing learning for simulations.

The comments given by previous reviewers are well addressed in making this manuscript with respect to their goal of making it.

Experimental design

The Comments given by previous reviewers are well addressed. Some demo applications are also given to give idea to the learners.
NO comments

Validity of the findings

The addressed comments from the previous version are seems to be good.

Additional comments

No comments

---

## Round 0.3 · accepted · Accept

The revised manuscript is recommended by the reviewers for acceptance.

Reviewer 1 ·

Basic reporting

Authors improved technical quality (presentation issues raised at the second review) of the reviewed paper. Authors answered on comments given at the second review. Therefore, the reviewed paper may be accepted for publication.

Experimental design

no comment

Validity of the findings

no comment

Additional comments

no comment